# LEARNING ENERGY DECOMPOSITIONS FOR PARTIAL INFERENCE IN GFLOWNETS

**Hyosoon Jang**[1]**, Minsu Kim**[2]**, Sungsoo Ahn**[1]
[1]POSTECH    [2]KAIST
{hsjang1205,sungsoo.ahn}@postech.ac.kr, min-su@kaist.ac.kr

## ABSTRACT

This paper studies generative flow networks (GFlowNets) to sample objects from the Boltzmann energy distribution via a sequence of actions. In particular, we focus on improving GFlowNet with *partial inference*: training flow functions with the evaluation of the intermediate states or transitions. To this end, the recently developed forward-looking GFlowNet reparameterizes the flow functions based on evaluating the energy of intermediate states. However, such an evaluation of intermediate energies may (i) be too expensive or impossible to evaluate and (ii) even provide misleading training signals under large energy fluctuations along the sequence of actions. To resolve this issue, we propose learning energy decompositions for GFlowNets (LED-GFN). Our main idea is to (i) decompose the energy of an object into learnable potential functions defined on state transitions and (ii) reparameterize the flow functions using the potential functions. In particular, to produce informative local credits, we propose to regularize the potential to change smoothly over the sequence of actions. It is also noteworthy that training GFlowNet with our learned potential can preserve the optimal policy. We empirically verify the superiority of LED-GFN in five problems including the generation of unstructured and maximum independent sets, molecular graphs, and RNA sequences.

## 1 INTRODUCTION

Generative Flow Networks (Bengio et al., 2021a, GFlowNets or GFNs) are frameworks to sample objects through a sequence of actions, e.g., iteratively adding nodes to a graph. Their key concept is training a policy that sequentially selects the actions to sample the object from the Boltzmann distribution (Boltzmann, 1868). Such concepts enable discovering diverse samples with low energies, i.e., high scores, as an alternative to reinforcement learning (RL)-based methods which tend to maximize the return of the sampled object (Silver et al., 2016; Sutton & Barto, 2018).

To sample from the Boltzmann distribution, GFlowNet trains the policy to assign action selection probability based on energy of terminal state (Bengio et al., 2021a;b; Malkin et al., 2022), e.g., a high probability to the action responsible for the low terminal energy. However, such training has fundamental limitations in credit assignment, as it is hard to identify the action responsible for terminal energy (Pan et al., 2023a). This limitation stems from solely relying on the terminal energy associated with multiple actions, lacking the information to identify the contribution of individual actions, akin to challenges in RL with sparse reward (Arjona-Medina et al., 2019; Ren et al., 2022).

An attractive paradigm to tackle this issue is *partial inference* (Pan et al., 2023a) that trains flow functions with local credits, e.g., evaluation of the intermediate states or transitions. Such local credit identifies individual action contributions to the terminal energy before reaching the terminal state. To this end, Pan et al. (2023a) proposed a forward-looking GFlowNet (FL-GFN), which assigns the local credit based on the energy of incomplete objects associated with intermediate states.

However, FL-GFN crucially relies on two assumptions that may not hold in practice. First, FL-GFN requires evaluating the energy of the intermediate state in the trajectories. However, the energy function can be expensive or even impossible to evaluate. Next, FL-GFN assumes the energy of intermediate states to provide useful hints for the terminal energy. However, this may not be true when the intermediate energy largely fluctuates along the sequence of states, e.g., low intermediate energies may lead to a terminal state with high energy. We illustrate such a pitfall in Figure 1.

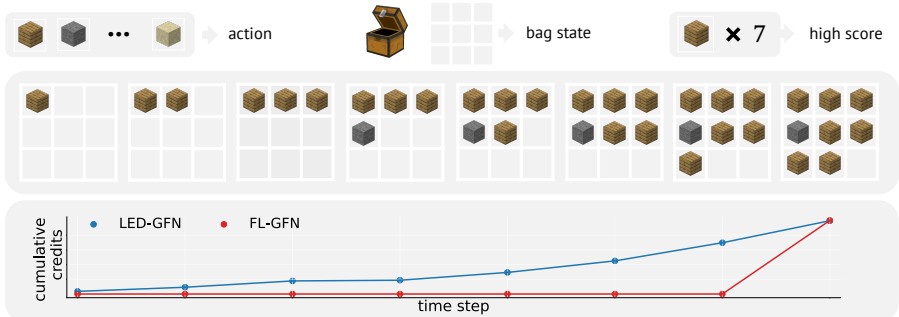

Figure 1: **The local credit evaluation in bag generation (Example 1)**. (**first row**) The task is to generate a bag of entities, where the seven same entities yield a high score. (**second row**) Left-to-right indicates the state transitions over a given trajectory. (**third row**) The energy-based evaluation fails to produce informative local credits since every intermediate state has zero energy, whereas our potential function produces informative credits by enforcing the potentials to be uniformly distributed.

**Contribution.** We propose learning energy decomposition for GFlowNet, coined LED-GFN. Our key idea is to perform partial inference by decomposing terminal state energy into a sum of learnable potentials associated with state transitions and use them as local credits. In particular, we show how to regularize the potential function to preserve the ground-truth terminal energy and minimize variance over the trajectory to yield informative potentials. Figure 1 highlights how LED-GFN provides informative local credits compared to the existing approach.

To be specific, our energy decomposition framework resembles the least square-based return decomposition for episodic reinforcement learning (Efroni et al., 2021; Ren et al., 2022). We parameterize potential function with a regression model that is constrained to be equal to the terminal energy when aggregated over the entire action sequence. We also regularize the potential function to minimize the variance along the trajectory, so that the potential function provides dense local credits in training GFlowNet. Such potentials associated with intermediate transitions provide informative signals, as each of them is enforced to be correlated with the terminal energies. The training of potential function is online, which uses samples collected during GFlowNet training.

We extensively validate LED-GFN on various tasks: set generation (Pan et al., 2023a), bag generation (Shen et al., 2023), molecular discovery (Bengio et al., 2021b), RNA sequence generation (Jain et al., 2022), and the maximum independent set problem (Zhang et al., 2023). We observe that LED-GFN (1) outperforms FL-GFN when the assumption of intermediate energy does not hold, (2) excels in practical domains compared to GFlowNets and RL-based baselines, and (3) achieves similar performance to FL-GFN even when intermediate energy provides the "ideal" local credit.

## 2 PRELIMINARIES

In this section, we describe generative flow networks (Bengio et al., 2021a, GFlowNets or GFNs) and their partial inference algorithm. We describe additional related works in Appendix A.

### 2.1 GFLOWNETS

GFlowNets sample from discrete space $\mathcal{X}$ through a sequence of actions from the action space $\mathcal{A}$ that make transitions in the state space $\mathcal{S}$. For each complete trajectory $\tau = (s_0, s_1, \ldots, s_T)$, the terminal state is the object $x = s_T \in \mathcal{X}$ to be generated. The state transitions are determined by the action sequence $(a_1, \ldots a_{T-1})$, e.g., $a_t$ determines $s_t \to s_{t+1}$. The policy $P_F(s'|s)$ selects the action $a$ to transition from the current state $s$ to the next state $s'$ and induces a distribution over the object $x$.

The main objective of GFlowNet is to train the policy $P_F(\cdot|\cdot)$ that samples objects from the Boltzmann distribution with respect to a given energy function $\mathcal{E}: \mathcal{X} \to \mathbb{R}$ as follows:

$$P_F^\top(x) \propto \exp(-\mathcal{E}(x)), \tag{1}$$

where $P_F^\top(x)$ is the distribution of sampling an object $x$ induced from marginalizing over the trajectories conditioned on $x = s_T$. We omit the temperature for simplicity. To this end, GFlowNet trains with auxiliary objectives based on state transition, trajectory, or sub-trajectory information.

**Detailed balance ([Bengio et al., 2021b](#), DB).** The DB utilizes the experience of state transitions to train GFlowNet. It trains the GFlowNet with a forward policy model $P_F(s'|s)$, a backward policy $P_B(s|s')$, and a state flow estimator $F(\cdot) : \mathcal{S} \to \mathbb{R}^+$ by minimizing the following loss function:

$$\mathcal{L}_{\text{DB}}(s, s') = \left(\log F(s) + \log P_F(s'|s) - \log F(s') - \log P_B(s|s')\right)^2,$$

where the flow $F(s)$ for the terminal state $s_T = x$ is defined to be identical to the exponent of the negative energy $\exp\left(-\mathcal{E}(x)\right)$, i.e., the score of the object.

**Trajectory balance ([Malkin et al., 2022](#), TB).** The TB aims to learn the policy faster by training on full trajectories. To this end, TB requires a forward policy model $P_F(s'|s)$, a backward policy $P_B(s|s')$, and a learnable scalar $Z$ to minimize the following loss function:

$$\mathcal{L}_{\text{TB}} = \left(\log Z + \sum_{t=0}^{T-1} \log P_F(s_{t+1}|s_t) - \mathcal{E}(x) - \sum_{t=0}^{T-1} \log P_B(s_t|s_{t+1})\right)^2.$$

This objective is resilient to the bias from inaccurate flow estimator $F(\cdot)$ used in DB, since it directly propagates the terminal energy to train on intermediate states. However, the TB suffers from the high variance of the objective over the collected trajectories ([Malkin et al., 2022](#)).

**Sub-trajectory balance ([Madan et al., 2023](#), subTB).** The subTB trains forward and backward policies $P_F(s'|s), P_B(s'|s)$, a flow function $F(\cdot)$, and a learnable scalar $Z$. It trains on flexible length of sub-trajectory $s_U \to s_{U+1} \cdots \to s_{U+L}$ to minimize the following loss function:

$$\mathcal{L}_{\text{subTB}} = \left(\log F(s_U) + \sum_{t=U}^{U+L-1} \log P_F(s_{t+1}|s_t) - \log F(s_{U+L}) - \sum_{t=U}^{U+L-1} \log P_B(s_t|s_{t+1})\right)^2.$$

In addition, one can consider the weighted average of $\mathcal{L}_{\text{subTB}}$ over all possible lengths $L = 1, \ldots, T$ of the sub-trajectories with the weight $\lambda^L$, which enables the interpolation between the DB and the TB ([Madan et al., 2023](#)). In this paper, we consider this approach as subTB for simplicity.

## 2.2 Partial inference in GFlowNets

The GFlowNet training is often challenged by limitations in credit assignment, i.e., identification and promotion of the action responsible for the observed low energy. This limitation stems from relying solely on the terminal state energy as the training signal. The terminal energy lacks information to identify the contribution of individual action, akin to how reinforcement learning with sparse reward suffers from credit assignment ([Arjona-Medina et al., 2019](#); [Ren et al., 2022](#)).

Partial inference is a promising paradigm to resolve this issue by learning from local credits, where [Pan et al. (2023a)](#) first incorporate this concept in training GFlowNets. Specifically, the partial inference aims to evaluate individual transitions or sub-trajectories, i.e., local credits, and provide informative training signals for identifying the specific contributions of actions. To this end, [Pan et al. (2023a)](#) proposed Forward-Looking GFlowNet (FL-GFN), which evaluates intermediate state energy as a local credit signal for partial inference.

**Forward-Looking GFlowNet ([Pan et al., 2023a](#), FL-GFN).** To enable partial inference, the FL-GFN defines a new training objective that incorporates an energy function $\mathcal{E} : \mathcal{X} \to \mathbb{R}$ for intermediate states. To this specific, FL-GFN modifies the DB as follows:

$$\mathcal{L}_{\text{FL}}(s, s') = (\log \tilde{F}(s) + \log P_F(s'|s) - \mathcal{E}(s) + \mathcal{E}(s') - \log \tilde{F}(s') - \log P_B(s|s'))^2, \quad (2)$$

where $\tilde{F}(s) = F(s) \exp\left(\mathcal{E}(s)\right)$ is the re-parameterized flow function and $\mathcal{E}(s') - \mathcal{E}(s)$ is the energy gain associated with the transition from $s$ to $s'$. Note that the energy function is defined only on the terminal state space $\mathcal{X}$; hence, FL-GFN assumes an extension of the energy function $\mathcal{E}$ to non terminal states, e.g., evaluating energy of an object associated with the intermediate state $s$. [Pan et al. (2023a)](#) shows that optimum of [Equation (2)](#) induces a policy $P_F(\cdot|\cdot)$ that samples from Boltzmann distribution. While [Equation (2)](#) is associated with DB, FL-GFN is also applicable to subTB with energy gains in the level of sub-trajectory.

Figure 2: **Illustration of energy decomposition for partial inference in GFlowNet.** LED-GFN enables partial inference with potentials which (a) approximate the object energy via summation, and (b) minimize variance along the action sequence.

## 3 LEARNING ENERGY DECOMPOSITION FOR GFLOWNET

While the FL-GFN is equipped with partial inference capabilities, it relies on the energy function to assign local credits, which can be expensive to evaluate, or lead to sub-optimal training signals (details are described in Section 3.1). In this paper, we propose learning energy decomposition for GFlowNets (LED-GFN) to achieve better partial inference. In what follows, we describe our motivation for better partial inference (Section 3.1) and the newly proposed LED-GFN (Section 3.2).

### 3.1 MOTIVATION FOR BETTER PARTIAL INFERENCE

Our motivation stems from the limitations of FL-GFN, which performs partial inference based on evaluating the energies of intermediate states with respect to a single transition (for DB) or sub-trajectory (for subTB). In particular, we are inspired by how the energy gain may yield a sub-optimal local credit signal due to the following pitfalls (see Figures 1 and 9(b)).

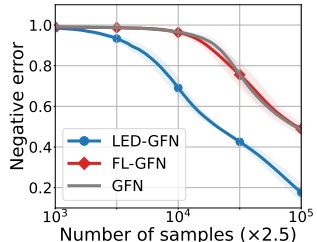

**A.** The energy evaluation for an incomplete object, i.e., intermediate state, can be non-trivial. In addition, the cost of energy evaluation can be expensive, which can bottleneck efficient training when called for all visited states.

**B.** The energy can exhibit sparse or high variance on intermediate states within a trajectory, even returning zero for most states, which is non-informative for partial inference.

Figure 3: Negative relative mean error ($\downarrow$) for estimating the true Boltzmann distribution on Example 1-type task (Bag).

We further provide a concrete example for the pitfall **B.** of FL-GFN, which is illustrated in Figure 1 and Figure 3[1] for both conceptual and empirical purposes, respectively:

**Example 1.** *Consider adding objects from* $\{A, B, C, D, E\}$ *to a bag with a maximum capacity of nine. Define the energy as* $-1$ *when the bag contains seven identical objects and* $0$ *otherwise.*

For Example 1, the intermediate energy (which is always $0$) does not provide information for the terminal energy. However, the number of the most frequent elements in the bag is informative even at intermediate states since greedily increasing the number leads to the best terminal state.

Our observation in Example 1 hints at the existence of a partial inference algorithm to provide better local credit signals. We aim to pursue this direction with a learning-based approach. That is, we parameterize the class of potential functions that decompose the terminal energy to provide local credit signals for GFlowNet training. Our key research direction is to understand about what conditions of the potential functions are informative for partial inference.

### 3.2 ALGORITHM DESCRIPTION

In this section, we describe our framework, coined learning energy decomposition for GFlowNet (LED-GFN), which facilitates partial inference using learned local credit. To this end, we propose to decompose the terminal energy into learnable potentials defined on state transitions. Similar to FL-GFN, we reparameterize the flow model with local credits, i.e., potentials. In contrast to FL-GFN,

---

[1]The detailed experiment settings are described in Section 4.1.

---

**Algorithm 1** Learning energy decomposition for GFlowNet

---

 1: Initialize the buffer $\mathcal{B}$, forward and backward policy $P_F, P_B$, state flow $\tilde{F}$, and model $\phi_\theta$.
 2: Update the model $\phi_\theta$ to minimize $\ell_{\text{ED}}$ if the generation trajectories are given in advance.
 3: **repeat**
 4:     Sample a batch of trajectories $\{\tau_b\}_{b=1}^{B_1}$ from forward policy $P_F$.
 5:     Update buffer $\mathcal{B} \leftarrow \mathcal{B} \cup \{\tau_b\}_{b=1}^{B_1}$.
 6:     **for** $n = 1, \ldots, N$ **do**                                   ▷ *Energy decomposition learning*
 7:         Sample a batch of trajectories $\{\tau_b\}_{b=1}^{B_2}$ from the buffer $\mathcal{B}$.
 8:         Update the model $\phi_\theta$ to minimize $\ell_{\text{LS}}$ with $\{\tau_b\}_{b=1}^{B_2}$.
 9:     **end for**
10:     Compute intermediate energy gains $\phi_\theta(s_i, a_i)$ for all $(s_i, a_i) \in \tau$.
11:     Update the GFlowNet $P_F, P_B, \tilde{F}$ to minimize $\mathcal{L}_{\text{LED}}$ with $\tau$ and $\phi_\theta(s_i, a_i)$ for all $(s_i, a_i) \in \tau$.
12: **until** converged

---

we optimize the local credits to enhance partial inference by minimizing the variance of potentials along the action sequence. See Figure 2 for an illustration of LED-GFN.

**Training with potential functions.** To be specific, we decompose the energy function $\mathcal{E}$ associated with the terminal state into learnable potential functions $\phi_\theta$ as follows:

$$\mathcal{E}(x) \approx \Phi_\theta(\tau) = \sum_{t=0}^{T-1} \phi_\theta(s_t \to s_{t+1}), \tag{3}$$

where $\tau = (s_0, s_1, \ldots, s_T)$, $x = s_T$, and the potential functions are defined on state transition $s_t \to s_{t+1}$. Similar to FL-GFN, we use the potential function to train the forward and backward GFlowNet policies $P_F, P_B$ and flow model $F$ to minimize the following loss:

$$\mathcal{L}_{\text{LED}}(s, s') = (\log \tilde{F}(s) + \log P_F(s'|s) + \phi_\theta(s \to s') - \log \tilde{F}(s') - \log P_B(s|s'))^2. \tag{4}$$

Given a sub-trajectory $(s_0, \ldots, s_u = s, s_{u+1} = s')$, one can derive this objective from Equation (2) by replacing $\mathcal{E}(s), \mathcal{E}(s')$ with $\sum_{t=0}^{u-1} \phi_\theta(s_t \to s_{t+1})$ and $\sum_{t=0}^{u} \phi_\theta(s_t \to s_{t+1})$, respectively. That is, it is evident that Equation (4) preserves the optimal policy of GFlowNet when $\mathcal{E}(x) = \Phi_\theta(\tau)$ is satisfied for all trajectories $\tau$ terminating with $x$.

Our objective becomes equivalent to that of FL-GFN when $\phi_\theta(s \to s') = \mathcal{E}(s') - \mathcal{E}(s)$, but our key idea is to *learn* the potential function $\phi_\theta$ instead of the energy gain which can be expensive and may exhibit sparsity or high variance, as pointed out in Section 3.1. Note that one can also introduce an approximation error $\mathcal{E}(x) - \Phi_\theta(\tau)$ as an additional correction term to preserve the optimal policy of GFlowNet even when the potential function $\phi_\theta$ is inaccurate. In Appendix B.1, we describe how LED-GFN consistently induces the optimal policy that samples from the Boltzmann distribution.

**Training potentials with squared loss.** In training the potential function, our key motivation is twofold: (a) accurately estimating the true energy through summation and (b) providing dense and informative training signals by minimizing variance along the action sequence.

To this end, given a trajectory $\tau = (s_0, \ldots, s_T = x)$, we train the potential functions $\phi_\theta$ to minimize the loss function for (a) achieving $\mathcal{E}(x) \approx \Phi_\theta(\tau)$ with (b) dropout-based regularization:

$$\ell_{\text{LS}}(\tau) = \mathbb{E}_{\boldsymbol{z} \sim \text{Bern}(\gamma)} \left[ \left( \frac{1}{T} \mathcal{E}(s_T) - \frac{1}{C} \sum_{t=0}^{T-1} z_t \phi_\theta(s_t \to s_{t+1}) \right)^2 \right]. \tag{5}$$

The $T$-length random vectors $\boldsymbol{z}$ promotes the dropout, where $z_t = 0$ sampled from the Bernoulli distribution with probability $1 - \gamma$. Dividing by $T$ and $C = \sum_{t=0}^{T-1} z_t$ aligns the scales to compensate for the scale reduction induced by dropout. When $\gamma = 1$ and the loss function is minimized, i.e., $\ell_{\text{LS}}(\tau) = 0$ for all $\tau$, the potential function decomposes the energy function without error. When $\gamma < 1$, dropout prevents heavy reliance on specific potentials to satisfy Equation (3), thereby reducing the variance and sparsity of the potentials along the action sequence.[2] Note that our intuition is

---

[2]Ren et al. (2022) provide a formal proof that Equation (5) serves as a surrogate objective to satisfy Equation (3) while reducing the variance and sparsity of the potentials along the action sequence.

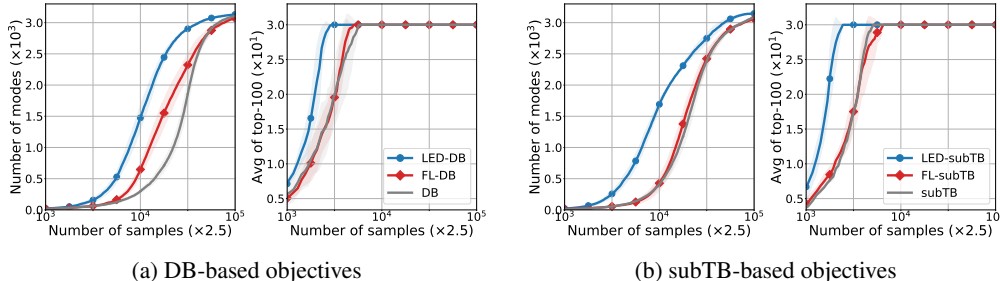

(a) DB-based objectives        (b) subTB-based objectives

Figure 4: **The performance on bag generation.** The solid line and shaded region represent the mean and standard deviation, respectively. The LED-GFN shows superiority on both DB and subTB.

similar to recent works on learning return decomposition to alleviate sparse reward problems in reinforcement learning (Arjona-Medina et al., 2019; Gangwani et al., 2020; Ren et al., 2022).

To train the potential function, we define its training as online learning within GFlowNet training, i.e., learning from trajectories obtained during GFlowNet training. We describe the overall algorithm in Algorithm 1. Such an alternative training of the potential function and the policy is similar to model-based reinforcement learning algorithms (Luo et al., 2018; Sun et al., 2018; Janner et al., 2019) for monotonic improvement of policies. Note that LED-GFN can also be implemented with subTB, where the details are described in Appendix B.2.

## 4 EXPERIMENT

We extensively evaluate LED-GFN on various domains, including bag generation (Shen et al., 2023), molecule generation (Bengio et al., 2021a), RNA sequence generation (Jain et al., 2022), set generation (Pan et al., 2023a), and the maximum independent set problem (Zhang et al., 2023). Following prior studies, we consider the number of modes, i.e., samples with energy lower than a specific threshold, and the average top-100 score as the base metrics, which are measured via samples collected during training. We report all the performance using three different random seeds.

### 4.1 BAG GENERATION

First, we consider a bag generation task (Shen et al., 2023). The action is adding an object from seven distinct entities to a bag with a maximum capacity of 15. The bag exhibits low energy when including seven or more repeated entities of the same type. In this task, we compare our method with GFN and FL-GFN. We consider both DB-based (Bengio et al., 2021b) and subTB-based (Malkin et al., 2022) implementations. The detailed experimental settings are described in Appendix C.

**Results.** We present the results in Figure 4. Here, one can observe that our method excels in bag generation compared to GFN and FL-GFN on both DB and subTB. In particular, FL-GFN fails to make improvements on the subTB-based implementation, since most states do not provide informative signals for partial inference (as illustrated in Figure 1). In contrast, LED-GFN consistently improves performance by producing informative potentials to enhance partial inference.

### 4.2 MOLECULE GENERATION

Next, we validate LED-GFN in a more practical domain: the molecule generation task (Bengio et al., 2021a). This task aims to find molecules with low binding energy to the soluble epoxide hydrolase protein. In this task, a molecule is generated by constructing junction trees (Jin et al., 2018), with the actions of adding molecular building blocks. The binding energy between the molecule and the target protein is computed using a pre-trained oracle (Bengio et al., 2021a).

In this experiment, we consider PPO (Schulman et al., 2017), and three GFN models: DB, TB, and subTB (Madan et al., 2023) as the baselines. Additionally, we compare our approach with GAFN (Pan et al., 2023b) and FL-GFN. For FL-GFN and LED-GFN, we consider a subTB-based implementation. The overall implementations and experimental settings follow prior studies (Bengio et al., 2021a; Pan et al., 2023a), which are described in Appendix C.

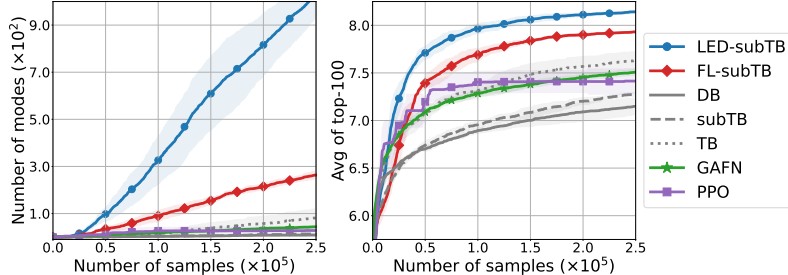

Figure 5: **The performance on molecule generation.** The solid line and shaded region represent the mean and standard deviation, respectively. The LED-GFN shows superiority compared to the considered baselines in generating diverse high reward molecules.

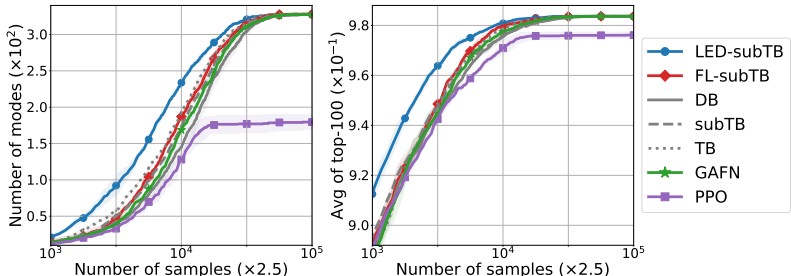

Figure 6: **The performance on RNA sequence generation.** The solid line and shaded region represent the mean and standard deviation, respectively. The LED-GFN shows superiority compared to the considered baselines in generating diverse high reward RNA sequences.

**Results.** The results are presented in Figure 5. One can see that LED-GFN outperforms the considered baselines in enhancing the average score of unique top-100 molecules and the number of modes found during training. These results highlight that LED-GFN is also beneficial for real-world generation problems with a large state space.

## 4.3 RNA SEQUENCE GENERATION

We also consider a RNA sequence generation task for discovering diverse and promising RNA sequences that bind to human transcription factors (Barrera et al., 2016; Trabucco et al., 2022; Jain et al., 2022). The action is appending or prepending an amino acid to the current sequence. The energy is pre-computed based on wet-lab measured DNA binding activity to Sine Oculis Homeobox Homolog 6 (Barrera et al., 2016). We consider the same baselines as in the molecule generation task. For FL-GFN and LED-GFN, we consider a subTB-based implementation.

**Results.** The results are presented in Figure 6. One can observe that LED-GFN outperforms the considered baselines. Furthermore, FL-GFN only makes minor differences compared to GFN, while LED-GFN makes noticeable improvements. These results highlight that energy-based partial inference can fail to improve performance in practical domains, while the potential learning-based approach consistently leads to improvements.

## 4.4 COMPARISON WITH IDEAL LOCAL CREDITS

In these experiments, we demonstrate that LED-GFN can achieve similar performance compared to FL-GFN, **even when the intermediate state energy is sufficient to identify the contribution of the action, i.e., ideal local credit** (Zhang et al., 2023). Note that this tasks are idealized, since designing such an energy function requires a complete understanding of the domain. Especially, we focus on two tasks: set generation (Pan et al., 2023a) and the maximum independent set problem (Zhang et al., 2023). For these tasks, we compare our method with GFN and FL-GFN.

**Set generation.** The set generation task is similar to the bag generation. The actions are adding an objects from 30 distinct objects to a set with a maximum capacity of 20. The energy is evaluated

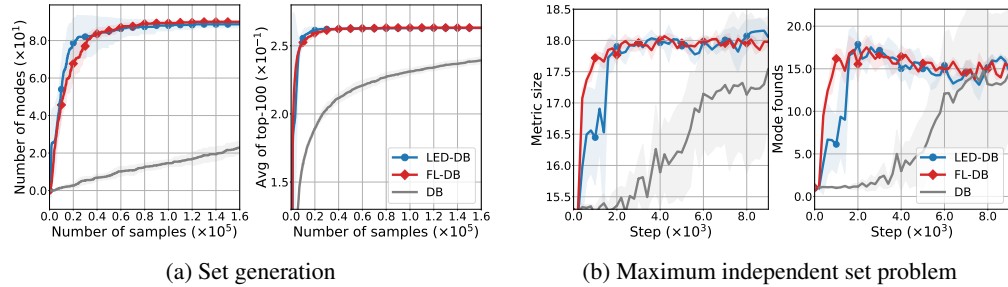

(a) Set generation       (b) Maximum independent set problem

Figure 7: **The performance comparison with ideal local credits.** The solid line and shaded region represent the mean and standard deviation, respectively. The LED-GFN shows similar performance to the FL-GFN, even when the intermediate energy is sufficient to identify the action contribution.

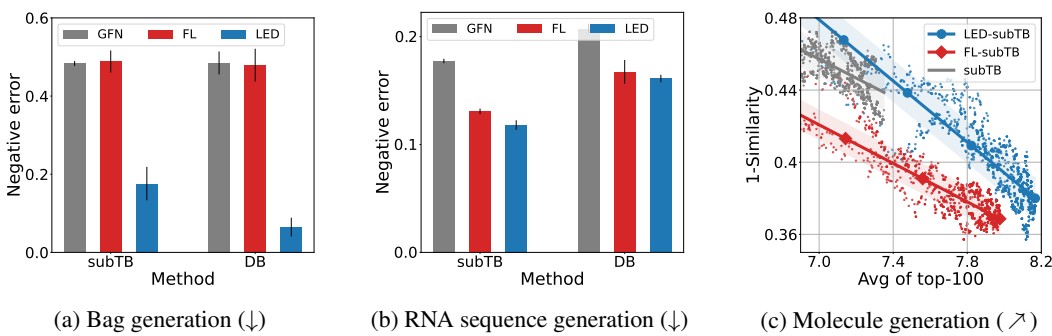

(a) Bag generation (↓)      (b) RNA sequence generation (↓)      (c) Molecule generation (↗)

Figure 8: **(a-b) The negative relative mean error comparison (lower is better).** Our LED-GFN better approximates the target distributions. **(c) The Tanimoto similarity with respect to the average scores (upper-right is better).** Our LED-GFN produces more diverse and promising molecules.

by accumulating the individual energy of each entity, so the intermediate energy gain has complete information to identify the contribution of each action (Pan et al., 2023a). We describe the detailed experiment settings in Appendix B.

**Maximum independent set problem.** This task aims to find the maximum independent set by selecting nodes, and the energy is evaluated based on the current size of the independent set (Zhang et al., 2023). Here, we compare the performance on validation graphs following Zhang et al. (2023). At each step, we sample 50 solutions for each validation graph to measure the average scores and the number of mode founds (greater than 18.5). The overall implementations and hyper-parameters follow prior studies (Zhang et al., 2023).

**Results.** As illustrated in Figure 7, one can see that our approach achieves similar performance to FL-GFN, even though the intermediate state energy provides ideal local credit for partial inference. These results highlight that the potentials of LED-GFN can be as informative as ideal local credit, which provides complete identification of the action contributions.

## 4.5 ABLATION STUDIES

**Goodness-of-fit to the true Boltzmann distribution.** We show how well our algorithm finds good samples sampled from the target distribution, i.e., Boltzmann distribution. Following Shen et al. (2023), we measure the relative mean error between the target distribution and empirical samples obtained during training. In Figures 8(a) and 8(b), one can observe that LED-GFN achieves a better approximation to target distribution compared to considered baselines.

**Diversity vs. high score.** Next, we further verify that our algorithm not only generates high-scoring samples but also diverse molecules. Specifically, we analyze the trade-off between the average score of the top-100 samples and the diversity these samples. To measure diversity, we compute the average pairwise Tanimoto similarity (Bajusz et al., 2015). In Figure 8(c), we illustrate the Tanimoto similarities with respect to the average of top-100 scores. Here, one can observe that LED-GFN achieves better diversity with respect to the average scores.

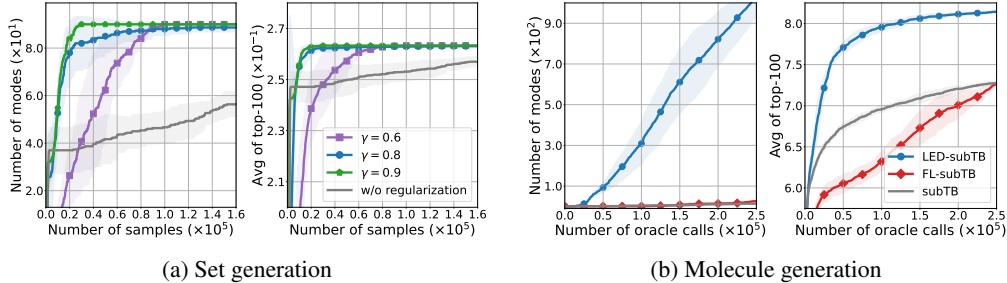

(a) Set generation             (b) Molecule generation

Figure 9: **(a) The benefits of regularizing variance of potentials.** The regularization improves the performance. **(b) The performance over the number of energy evaluation.** The FL-GFN can not make improvements with respect to the number of energy evaluation.

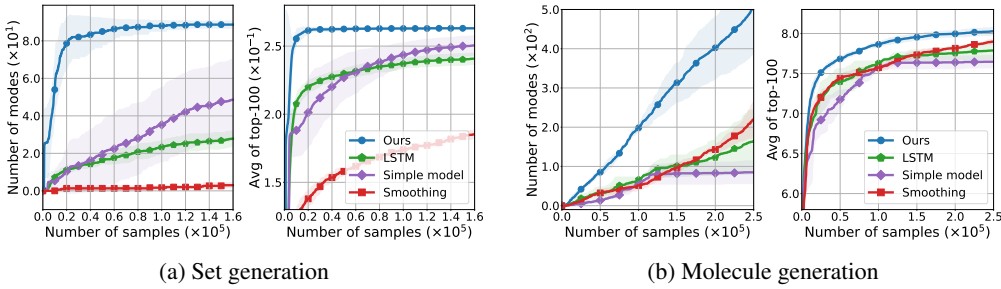

(a) Set generation             (b) Molecule generation

Figure 10: **Comparison between energy decomposition methods for partial inference in GFlowNets.** Our energy decomposition learning scheme shows most promising results.

**Regularized vs. non-regularized potentials.** We also analyze how reducing the variance of potentials benefits the improvement in performance. In the set generation, we compare LED-DB with various dropout rates $1 - \gamma = 0.1, 0.2, 0.4$ and its counterpart without regularization $1 - \gamma = 0$, in Equation (5). In Figure 9(a), one can see that applying regularization yields more promising performance, although too high a dropout rate may result in inefficiency.

**Number of energy evaluation vs. performance** We analyze the performance with respect to the number of energy evaluation, which can be expensive. In Figure 9(b), one can see that FL-GFN can not improve performance since it requires evaluating the energy for every visited states. In contrast, LED-GFN uses a potential function without energy evaluation for intermediate states.

**Comparison with alternative energy decomposition methods.** We further investigate the following alternative energy decomposition schemes to enable the partial inference in GFlowNets.

- One may train a **simple model** $\phi_\theta : \mathcal{X} \to \mathbb{R}$ to predict the terminal energy, and utilize it to compute potentials $\phi_\theta(s \to s') = \phi_\theta(s') - \phi_\theta(s)$. This approach can be interpreted as extension of proxy model-based GFlowNet (Jain et al., 2022) for partial inference.
- Based on the **LSTM**-based decomposition method (Arjona-Medina et al., 2019), one can design the potential $\phi_\theta(s_t \to s_{t+1})$ as the difference between two subsequent predictions for $(s_0, a_0, \ldots, s_t, a_t)$ and $(s_0, a_0, \ldots, s_{t+1}, a_{t+1})$ using an LSTM.
- Based on the return **smoothing** over the trajectory (Gangwani et al., 2020), one can simply set $\phi_\theta(s_t \to s_{t+1}) = \mathcal{E}(s_T)/T$ for a given trajectory $\tau$ at each step.

In Figure 10, we compare each method in molecule and set generation tasks with a DB-based implementation. Here, one can see that the least square-based approach shows the most competitive performance due to its capabilities in producing dense and informative potentials.

## 5 CONCLUSION

In this paper, we propose learning energy decomposition for GFlowNets (LED-GFN). Experiments on various domains show that LED-GFN is promising compared to existing partial inference methods for GFlowNet. An interesting avenue for future work is developing new partial inference techniques using learnable local credit, or theoretically analyzing learned potentials and optimal local credits.

**Reproducibility.** We describe experimental details in Appendix C, which provides the base implementation references, environments, and detailed hyper-parameters. In the supplementary materials, we also include the codes for molecule generation tasks based on the official implementation codes of the prior study (Pan et al., 2023a).

## ACKNOWLEDGEMENTS

This work partly was supported by Institute of Information & communications Technology Planning & Evaluation (IITP) grant funded by the Korea government(MSIT) (No. IITP-2019-0-01906, Artificial Intelligence Graduate School Program(POSTECH)), the National Research Foundation of Korea(NRF) grant funded by the Korea government(MSIT) (No. 2022R1C1C1013366), and Basic Science Research Program through the National Research Foundation of Korea(NRF) funded by the Ministry of Education(2022R1A6A1A0305295413).

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

## A    RELATED WORKS

**Generative augmented flow network (Pan et al., 2023b, GAFN).** The GAFN is a learning framework that incorporates intermediate rewards for exploration purposes. Specifically, GAFN measures the novelty score of a given state, which provides an intrinsic signal to facilitate better exploration towards unvisited states. To compute the novelty score, this method also incorporates online training of random network distillation (Burda et al., 2019), which assigns lower scores to unseen states compared to more frequently observed states.

**Proxy model-based GFlowNet.** Jain et al. (2022) propose proxy model-based training of GFlowNet for discovering diverse and promising biological sequences. They train a proxy model of the energy function to mitigate the expensive cost of evaluating biological sequences, such as wet-lab evaluation. Additionally, they introduce an active learning algorithm for the model-based GFlowNet, leveraging the epistemic uncertainty estimation of the model to improve exploration.

**Return decomposition learning.** Our LED-GFN approach is inspired by return decomposition learning of reinforcement learning in sparse reward settings. Their goal is to decompose the return into step-wise dense reward signals (Arjona-Medina et al., 2019; Gangwani et al., 2020; Ren et al., 2022). They have studied various return decomposition methods. First, Arjona-Medina et al. (2019) utilize an LSTM-based model to produce step-wise proxy rewards. Next, Gangwani et al. (2020) propose a simple approach that uniformly redistributes the terminal reward over the trajectory. Ren et al. (2022) train a proxy reward function using randomized return decomposition learning which is contrained to produces dense and informative proxy rewards.

# B    DETAILS OF LED-GFN

## B.1    PRESERVING OPTIMAL POLICY OF GFLOWNET

Although the potential function is inaccurate, we show that optimum of $\mathcal{L}_{\text{LED}}$ can induce an optimal policy that samples from Boltzmann distribution. We give a simple proof by reduction to the optimum of DB objective (Bengio et al., 2021b). Suppose the parameterization $\phi_\theta(s \to s') = \phi_\theta(s') - \phi_\theta(s)$, and $\log \hat{F}(s) = -\phi_\theta(s) + \log \tilde{F}(s)$. Then, we can reformulate $\mathcal{L}_{\text{LED}}$ as follows:

$$\mathcal{L}_{\text{LED}}(s, s') = \Big( \log \hat{F}(s) + \log P_F(s'|s) - \log \hat{F}(s') - \log P_B(s|s') \Big)^2 .$$

If we add an additional correction term $-\mathcal{E}(x) + \Phi_\theta(\tau) = -\mathcal{E}(x) + \phi_\theta(x)$ to the terminal flow such that $\log \hat{F}(x) = -\mathcal{E}(x)$, this objective becomes to be equivalent to the reparameterization of DB, where the optimum induces a policy sampling from a Boltzmann distribution (Bengio et al., 2021b). Therefore, the optimum of LED-GFN can still induce the policy that samples from the Boltzmann distribution. We refer to this correction-based approach as LED-GFN*.

However, our implementation follows a prior study in return decomposition learning (Ren et al., 2022), which uniformly redistributes the decomposition error over the transitions within the given trajectory (we denote this approach as LED-GFN in experiments). In Figure 11, we empirically observe that this approach further improves the training of GFlowNets. We assume that uniformly redistributed decomposition error partially provides more dense and informative local credit signals correlated with future energy.

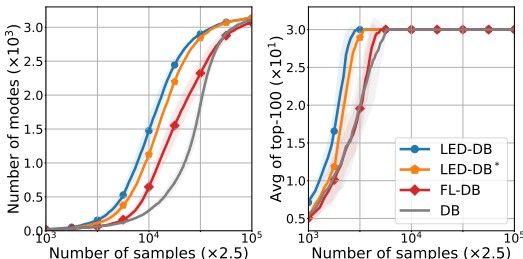

Figure 11: The performance on bag generation.

## B.2    TRAINING ON SUBTB

The LED-GFN can also be implemented on subTB by incorporating the potentials within sub-trajectories. To this specific, one can modify subTB as follows:

$$\mathcal{L}_{\text{LED-subTB}} = \Big( \log \tilde{F}(s_U) + \sum_{t=U}^{U+L-1} \log P_F(s_{t+1}|s_t) + \sum_{t=U}^{U+L-1} \phi_\theta(s_t \to s_{t+1})$$
$$- \log \tilde{F}(s_{U+L-1}) - \sum_{t=U}^{U+L-1} \log P_B(s_t|s_{t+1}) \Big)^2 ,$$

which is based on a sub-trajectory $s_U \to s_{U+1} \cdots \to s_{U+L-1}$. This objective is equivalent to FL-GFN on the subTB when $\phi_\theta(s \to s') = \mathcal{E}(s') - \mathcal{E}(s)$ (Pan et al., 2023a), but we replace it with the potentials for better credit assignment. Note that we actually consider the weighted average of $\mathcal{L}_{\text{LED-subTB}}$ over all possible lengths $L = 1, \ldots, T$ of the sub-trajectories with the weight $\gamma^L$.

## C EXPERIMENTAL DETAILS

In all experiments, we design the potential functions to have the same architecture as the flow model, e.g., a feedforward network with same hidden dimensions, where the input dimension is extended to consider the transition $s_t \to s_{t+1}$. In Equation (5), we set the dropout rate $\gamma$ as $10\%$ for tasks with a trajectory length less than $10$ and $20\%$ for others.

**Bag generation (Shen et al., 2023).** The experiment settings, implementations, and hyper-parameters are based on prior studies (Shen et al., 2023). The bag generation task is to generate a bag with a maximum capacity of $15$. There are seven types of entities, and each action includes one of them in the current bag. If it contains seven or more repeats of any items, it has a reward $10$ with $75\%$ chance, and $30$ otherwise. The threshold for determining the mode is $30$.

In each round, we generate $B_1 = 32$ bags from the policy. The GFlowNet model consists of two hidden layers with $16$ hidden dimensions, which is trained with a learning rate of $1\mathrm{e}{-4}$. We use an exploration epsilon of $0.01$. In this task, we incorporate a buffer for energy decomposition learning. The mini-batch size $B_2$ is same as $B_1$. We set the number of iterations in energy decomposition learning $N = 8$ for each round. Note that reducing $N$ still leads to promising results compared to baseline.

**Molecule generation (Bengio et al., 2021a).** The experiment settings, implementations, and hyper-parameters are based on prior studies (Bengio et al., 2021a; Pan et al., 2023a). The maximum trajectory length is $8$, with the number of actions varying between around $100$ and $2000$ which is depending on the state. The threshold for determining the mode is $7.5$.

In each round, we generate four molecules, i.e., $B_1 = 4$. The model consists of Message Passing Neural Networks (Gilmer et al., 2017) with ten convolution steps and $256$ hidden dimensions, which is trained with a learning rate of $5\mathrm{e}{-4}$. We rescale the reward so that the maximum reward is close to one, and the exponent of it is set to $8.0$. We use an exploration epsilon of $0.05$. In energy decomposition learning, we do not use a buffer and immediately use the molecules that are sampled in each round. For PPO, we set the entropy coefficient to $1\mathrm{e}{-4}$ and do not apply the reward exponent because it causes a gradient exploding.

**RNA sequence generation (Shen et al., 2023).** The experiment settings, implementations, and hyper-parameters are based on prior studies (Shen et al., 2023). The action is defined as prepending or appending an amino acid to the current sequence. The maximum length is $8$ and the number of actions is $4$. The mode is determined based on whether it is included in a predefined set of promising RNA sequences (Shen et al., 2023).

In each round, we generate $B_1 = 16$ sequences. The GFlowNet model consists of two hidden layers with $128$ hidden dimensions, which is trained with a learning rate of $1\mathrm{e}{-4}$. The reward exponent is set to $3.0$. We use an exploration epsilon of $0.01$. In this task, we incorporate a buffer for energy decomposition learning. The hyper-parameters for energy decomposition learning are the same as that of bag generation. For PPO, we set the entropy coefficient to $1\mathrm{e}{-2}$.

**Set generation (Pan et al., 2023a).** The experiment settings, implementations, and hyper-parameters are based on prior studies (Pan et al., 2023a). The set generation task is to generate a set that involves $20$ entities. There are $30$ types of entities, where each action includes one of them in the current set. We set the threshold for determining the mode as $0.25$.

In each round, we generate $B_1 = 16$ sets. The GFlowNet model consists of two hidden layers with $256$ hidden dimensions, which is trained with a learning rate of $0.001$. The hyper-parameters for energy decomposition learning are the same as that of molecule generation.

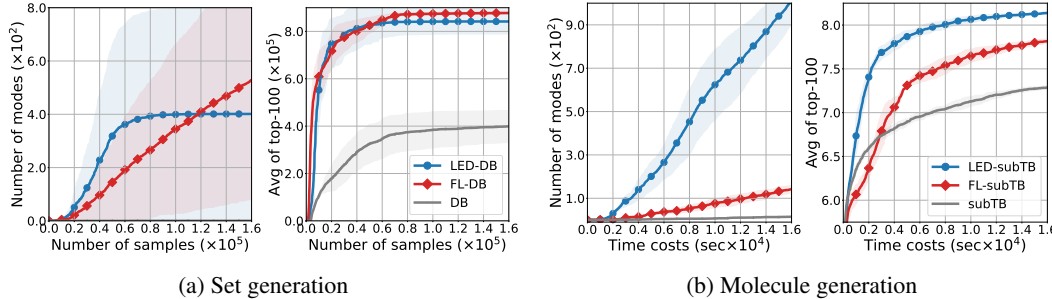

(a) Set generation          (b) Molecule generation

Figure 12: **(a) Large set generation.** The partial inference significantly improves the performance. **(b) The performance with respect to the time costs.** The LED-GFN shows most promising results.

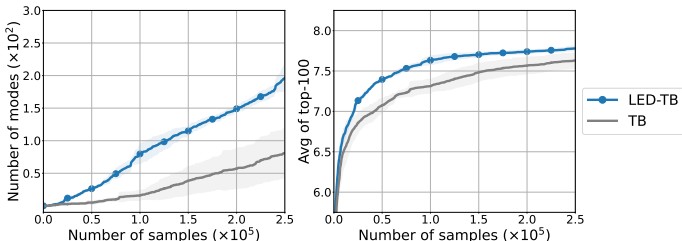

Figure 13: **The performance on molecular generation.** The energy decomposition learning also benefits the TB-based objective.

## D   ADDITIONAL ABLATION STUDIES

### D.1   LARGE SET GENERATION

We conduct additional experiments with relatively long trajectories: the set generation with the size of 80 where the actions are adding an objects from 100 distinct objects (Pan et al., 2023a). This task is more likely to make the local credit assignment more challenging as a lot of actions are associated with the terminal energy. Note that this setting is an idealized setting for FL-GFN, where intermediate energy gain has complete information to identify the contribution of each action (setting discussed in Section 4.4). In Figure 12(a), one can observe that LED-GFN (1) significantly outperforms the GFN and (2) shows similar performance compared to FL-GFN with ideal local credits. This result highlights that learned potentials can be informative as ideal local credits.

### D.2   TIME COST ANALYSIS

We also analyze time costs. Especially, we consider molecular generation tasks and compare LED-GFN with GFN and FL-GFN on subTB-based objectives. We use a single GPU of NVIDIA A5000 for this experiment. In Table Table 1, one can see that LED-GFN only incurs small overheads (approximately 10%) compared to GFN. In Figure Figure 12(b), we also compare the performance with respect to the time costs. One can see that LED-GFN achieves the most promising results with respect to the time costs.

Table 1: **Time costs (sec) analysis.** The LED-GFN does not incur significant overheads.

| Method | Time cost |
|---|---|
| subTB | 5.80 ($-$) |
| FL-subTB | 11.43 ($\uparrow$ 5.63) |
| LED-subTB | 6.61 ($\uparrow$ 0.81) |

### D.3   TB-BASED OBJECTIVE

We also combine energy decomposition learning to TB-based objectives by reparameterizing the logits of the policy with the potential $\phi_\theta(s \rightarrow s')$. In Figure Figure 13, one can see that our energy decomposition even improves on the TB. We assume that local credits also benefit the TB since they support assigning a high probability to the action responsible for the low terminal energy.

