# OpenReview forum: "Learning Energy Decompositions for Partial Inference in GFlowNets"
_ICLR.cc/2024/Conference — ICLR 2024 oral_

### Official Review · Reviewer_qpM8 · 2023-10-30

**Soundness:** 3 good
**Presentation:** 3 good
**Contribution:** 3 good
**Rating:** 8
**Confidence:** 5

**Summary:**

This paper investigates generative flow networks (GFlowNets), which sample a composite object via a sequence of constructive steps, with a probability proportional to a reward function. In contrast to prior training objectives for GFlowNets, which mainly focus on learning from complete trajectories, Looking-Forward GFlowNets (FL-GFlowNets, as introduced by Pan et al.) take advantage of the computability of intermediate rewards or energies and employ an additive energy factorization to facilitate learning from incomplete trajectories. However, the authors argue that there are two limitations: 1) intermediate rewards are too expensive to be evaluated, e.g., frequent evaluation over partially-constructed molecules at all intermediate steps can indeed be time-consuming, especially in the context of chemical synthesis or molecular design; 2) the significant variability in intermediate rewards along the entire trajectory, e.g., rewards are low or zero at the beginning, and experience a sudden surge at the end (as illustrated in Figure 1), thus leading to less informative intermediate signals. To this end, inspired by the reward decomposition method in Ren et al., the authors propose to learn a potential function that is additive over transitions. This is done by minimizing the least square loss between $R(s_{n})$ and a summation over potential functions that incorporates a dropout-based technique, i.e., $\sum_{t=0}^{n-1} z_{t} \phi_{\theta} (s_{t} \rightarrow s_{t+1})$, together with some scaling coefficients. The learned potential function can be directly incorpoated into FL-GFlowNets, thus leading to LED-GFlowNets. Experimental results on several datasets show the effectiveness of LED-GFlowNets.

**Strengths:**

### Motivation:

A major limitation of GFlowNets is that they suffer from inefficient credit assignment when trajectories are long, as the learning signal only comes from the terminal reward (episodic setting). This is where partial inference comes in - the learning signal is from partial reward signals. To this end, FL-GFlowNets take advantage of intermediate rewards and an additive energy factorization over transitions to facilitate learning from incomplete trajectories. However, there might be two limitations: 1) intermediate rewards are too expensive to be evaluated; 2) the significant variability in intermediate rewards along the entire trajectory. These motivate LED-GFlowNets.


### Originality:

The proposed method builds on two existing works - reward decomposition [Ren et al.] and FL-GFlowNets [Pan et al.]. The authors extend the reward decomposition method by introducing a dropout-based regularizer to reduce variance and promote sparsity. The learnable potential function $\phi_{\theta} (s_{t} \rightarrow s_{t+1})$ can be directly used in FL-DB or FL-SubTB. The combination of existing ideas shows good performance on various tasks. The originality should be ok.


### Clarity:

The paper is well-organized and easy to follow, making it accessible to readers.

**Weaknesses:**

Please see the following questions.

**Questions:**

### Method:

- The paper considers GFlowNets whose reward function $R(s_{n})$ corresponds to a potential function that is additive over transitions, i.e., $- \log R(s_{n}) = \mathcal{E}(s_{n}) = \sum_{t=0}^{n-1} \phi_{\theta} (s_{t} \rightarrow s_{t+1})$. This makes sense for set GFlowNets, where $s_{n}$ contains information about all the transitions, but not about their order, as $s_{n} = x$ is the set of elements that have been added at each transition. Is it also applicable for the tasks where order might matter?

- In terms of Figure 2,
   - Why do we want to have (b) for potentials --> approximately same energies to minimize variance (more smooth transitions)? In this case, we just simply set $\frac{1}{3}$ if $\mathcal{E} = 1$?
   - When $e^{- \mathcal{E}(s)} = e^{- \mathcal{E}(s^{\prime})} \Rightarrow \mathcal{E}(s \rightarrow s^{\prime}) = 0$, it reduces to the DB constraint, such that we cannot take advantage of intermediate rewards or energies to learn from incomplete trajectories? I conjecture this might be a common case in many tasks?

- In GFlowNet settings, we hope to achieve a transition probability distribution: $P_{F}(s_{t+1} | s_{t}) \propto F(s_{t} \rightarrow s_{t+1})$. I am curious - since we have $- \log R(s_{n}) = \mathcal{E}(s_{n}) = \sum_{t=0}^{n-1} \phi_{\theta}(s_{t} \rightarrow s_{t+1})$, then we might be able to learn $P_{F}(s_{t+1} | s_{t}) \propto \phi_{\theta}(s_{t} \rightarrow s_{t+1})$? With such policy, can we achieve - sampling $x$ with probability proportional to $R(x)$? If not, with GFlowNet training objectives, the learned policy would be modified accordingly?

### Experiment:

- In terms of Figure 9(b), how to understand number of calls? Assume we have 16 trajectories now, LED-GFlowNets compute 16 terminal rewards; while FL-GFlowNets need to compute all intermediate and terminal rewards (should be $\sum_{i=1}^{16} n_{i}$, where $n_{i}$ is the number of states, except $s_{0}$, in the ${i}$-th trajectory?). Thus, 16 calls vs. $\sum_{i=1}^{16} n_{i}$ calls? But, I think we should have more than 16 calls for LED-GFlowNets, as we need to train the potential function.

- For experimental details, 'We set the dropout probability as 10% for tasks with a trajectory length less than 10 and 20% for others.' -- Did you try other proportions? How does $\lambda$ affect the performance? Abalation studies on $\lambda$ for Figure 9 (a) are missing.


### Some writing issues, typos and inconsistencies:

1) $\log$ is missing for $P_B$ regarding the DB loss, as well as in the Appendix.

2) As far as I understand, it should be --> SubTB($\lambda$) [Madan et al.] is practically useful interpolation between TB and DB losses?

3) Below equation (4), .... by replacing $\mathcal{E}(s)$, $\mathcal{E}(s^{\prime})$ with --> should be $\sum_{t=0}^{u} \phi_{\theta}(s_{t} \rightarrow s_{t+1})$ and $\sum_{t=0}^{u+1} \phi_{\theta}(s_{t} \rightarrow s_{t+1})$, respectively.?

4) Figure 4: SubTB-based objectives --> subTB-based objectives

5) For Figure 4, it is clear that DB or subTB is considered. But for Figure 5 & 6, it's just LED-GFN or FL-GFN. Which training objective are you using? (though 'For FL-GFN and LED-GFN, we consider a subTB-based implementation.' is mentioned for molecule generation). Maybe just use LED-subTB for the figures? Try to find a way to avoid such confusion.

---

**Update after rebuttal**

I raised the score: 6 --> 8.

---

> ### Author Response · Authors · 2023-11-15
>
> Dear reviewer qpM8,
>
> We express our deep appreciation for your time and insightful comments. In our updated manuscript, we highlight the changes in $\color{blue}{\text{blue}}$.
>
> In what follows, we address your comments one by one.
>
> ---
>
> **Q1-Method. Is LED-GFN also applicable for the tasks where the order of actions might matter?**
>
> Yes, to the best of our knowledge, all the GFlowNet frameworks (including LED-GFN) are applicable to scenarios where the order of actions matters. To this end, one can define the state to incorporate the order of actions, e.g., a state $s_{t}$ information may include the feature $[(a_0,0),\ldots,(a_{t-1},{t-1})]$ to ensure the order of actions-dependent GFlowNets.
>
>
> ---
>
> **Q2-1-Method. Why do we want to smooth potentials? In this case, can we just simply set each potential to $\frac{\mathcal{E}}{T}$?**
>
> As mentioned in **Section 3.1**, we want to smooth potentials to prevent heavily relying on a specific transition, e.g., $\mathcal{E}(s_T)=\phi_{\theta}(s_{T-1}\rightarrow s_{T})$. However, we do not set $\phi_{\theta}(s \rightarrow s') = \mathcal{E}(s_T)/T$ for a specific trajectory, as the potential is associated with *multiple trajectories* $\\{\tau |  (s\rightarrow s') \in \tau\\}$. In other words, the potential $\phi_{\theta}(s \rightarrow s')$ should be designed to consider:
> \begin{equation*}
>     \mathcal{E}(s_{T})\approx\sum_{(s_{t}, s_{t+1}) \in \tau}\phi_{\theta}(s_{t}\rightarrow s_{t+1})\quad \text{for all } \tau=(s_{0},\ldots,s_{T}) \in \\{ \tau | (s\rightarrow s') \in \tau \\}.
> \end{equation*}
> Then, one can observe that the condition $\phi_{\theta}(s \rightarrow s') = \mathcal{E}(s_T)/T$ for a specific trajectory would not be optimal for another trajectory. In **Figure 10** of our updated manuscript, we add such a baseline, i.e., set $\phi_{\theta}(s \rightarrow s') = \mathcal{E}(s_T)/T$ for a given trajectory at each step. One can see that our learning method still exhibits the most competitive performance.
>
> ---
>
> **Q2-2-Method. When $\mathcal{E}(s\rightarrow s')=0$, FL-GFN reduces to the DB, but is it common case in many tasks?**
>
> Yes, it can be a common case in real-world settings. Especially, when the energy evaluation of the intermediate state may be impossible, e.g., wet-lab evaluation for incomplete molecules, the change of energy would be evaluated as zero.
>
> ---
>
> **Q3-Method. LED-GFN might learn $P_F(s_{t+1}|s_t)\propto \phi_{\theta}(s_t\rightarrow s_{t+1})$, but can LED-GFN sample $x$ with probability proportional to $\exp{(-\mathcal{E}(x))}$?**
>
> We would like to clarify that LED-GFN learns a policy to be $P_F(s_{t+1}|s_t)\propto {F}(s_t\rightarrow s_{t+1})$ since our objective (Equation (4)) just reparameterizes the flow of the GFN with the potential: $\log{F}(s_t)=\log\tilde{F}(s_t)-\sum_{u=0}^{t-1}{\phi_{\theta}(s_{u}\rightarrow s_{u+1})}$. Hence, LED-GFN can sample $x$ with probability proportional to $\exp{(-\mathcal{E}(x))}$.
>
> ---
>
> **Q1-Experiment. In Figure 9(b), what is the number of calls, and why is the number of calls for LED-GFN the same as that for GFN?**
>
> The number of calls is the number of energy evaluations. LED-GFN has the same number of calls as the GFN since the training of the potential function does not require additional energy evaluation, as it learns from the trajectories and energies that have already been evaluated during GFN training (mentioned in the last paragraph of **Section 3.2**).
>
> ---
>
> **Q2-Experiment. Ablation studies on various dropout rates $1-\gamma$ for Figure 9(a) are missing. How does the dropout rate affect performance?**
>
> To address your comment, we incorporate experiments with dropout rates $10\\%$, $20\\%$, and $40\\%$ for the set generation and incorporate the results in **Figure 9(a)** of our updated manuscript. One can see that (1) applying dropout benefits credit assignment, (2) but too high a dropout rate may result in inefficient training.
>
> ---
>
> **Additional minor issues in writing.**
>
> Thank you for pointing this out! We update our manuscript to follow your valuable suggestions.
>
> - We add $\log$ for $P_B$ in the DB objective.
> - We clarify that we actually consider the subTB($\lambda$).
> - We modify $\sum_{t=0}^{u}\phi(s_{t})$ and $\sum_{t=0}^{u+1}\phi(s_{t})$ to $\sum_{t=0}^{u-1}\phi(s_{t}\rightarrow s_{t+1})$ and $\sum_{t=0}^{u}\phi(s_{t}\rightarrow s_{t+1})$, respectively.
> - We fix 'SubTB' to 'subTB' in Figure 4.
> - We fix 'FL-GFN' and 'LED-GFN' in Figure (5) and (6) to 'FL-subTB' and 'LED-subTB', respectively.

---

> > ### Comment · Reviewer_qpM8 · 2023-11-19
> > **Thank you for your response!**
> >
> > Thank you for the detailed response. I appreciate your efforts to improve the paper and add the new experiments.
> >
> > ---
> >
> > **Q2-1-Method**
> >
> > According to Figure 11, it's surprising to me that GFlowNets with uniformly redistributed energy decomposition can still outperform FL-GFlowNets and vanilla GFlowNets.
> >
> > ---
> >
> > **Q3-Method**
> >
> > Since we have $- \log R(s_{n}) = \mathcal{E}(s_{n}) = \sum_{t=0}^{n-1} \phi_{\theta}(s_{t} \rightarrow s_{t+1})$, we might be able to learn $P_{F}(s_{t+1} | s_{t}) \propto \phi_{\theta}(s_{t} \rightarrow s_{t+1})$, as we know $\phi_{\theta}(s_{t} \rightarrow s^{\prime}), s^{\prime} \in Child(s_{t})$, or equivalently probability. Such energy factorization tells us the information about all the transitions, but not about their order. I conjecture that it might not be able to achieve $P_{T}(x) \propto R(x)$.
> >
> > ---
> >
> > **Q1-Experiment**
> >
> > I didn't think I express my confusion clearly. The number of calls for FL-GFlowNets and GFlowNets differs, as FL-GFlowNets need to evaluate all intermediate energies.
> >
> > ---
> >
> > Some writing issues after I read your updated manuscript:
> >
> > - The subTB trains forward and backward policies $P_{F}(s^{\prime} | s)$, $P_{B}(s^{\prime} | s)$ and a learnable scalar $Z$ similar to the TB. --> should be 1) $P_{B}(s | s^{\prime})$; 2) a flow function $F(\cdot)$ similar to the DB?
> >
> > - 2.2 Partial Inference for GFlowNets --> Partial Inference in GFlowNets? as the title was modified.
> >
> > - Below equation (4), the expressions highlighted in blue are missing the parameter $\theta$? --> $\phi_{\theta}()$
> >
> > - In terms of Figure 11, LED-DB* should correspond to GFlowNets with uniformly redistributed energy decomposition? If yes, maybe add some descriptions.

---

> > > ### Author Response · Authors · 2023-11-20
> > >
> > > Dear reviewer qpM8,
> > >
> > > Thank you for the response. We appreciate your comments to improve our paper!
> > >
> > > We would like to clarify the following points to address your additional comments.
> > >
> > > ---
> > >
> > > **Q2-1-Method-1. According to Figure 11, it's surprising that uniformly redistributed energy decomposition can outperform GFN and FL-GFN.**
> > >
> > > We would like to clarify that the performance of uniformly redistributed energy is presented as **Smoothing** in **Figure 10** as an alternative method for defining potentials. The corresponding information is provided in the last bullet in the last paragraph of **Section 4.5**.
> > >
> > > One can see that GFN with uniformly redistributed energy fails in the set generation **Figure 10(a)** while showing good performance for molecular generation **Figure 10(b)**.
> > >
> > > ---
> > >
> > > **Q3-Method-1. If we learn $P_F(s_{t+1}|s_t)\propto \exp{(-\phi_{\theta}(s_t\rightarrow s_{t+1}))}$ instead of training with GFNs, is it possible to learn $P_F^{\top}(x)\propto \exp{(-\mathcal{E}(x))}$?**
> > >
> > > We first apologize for misunderstanding your original question **Q3-Method**. In addition to your conjecture, we provide a clear response in what follows.
> > >
> > > Although we learn $P_F(s_{t+1}|s_t)\propto \exp{(-\phi_{\theta}(s_t\rightarrow s_{t+1}))}$ instead of LED-GFN, we are unable to learn $P_F^{\top}(x)\propto \exp{(-\mathcal{E}(x))}$. Because, $P_F(s_{t+1}|s_t)\propto \exp{(-\phi_{\theta}(s_t\rightarrow s_{t+1}))}$ just consider the immediate potentials, but it can not consider the future transition information.
> > >
> > > For a counter-example, let's consider there are only two trajectories $\tau=(s_0,s_1)$ and $\tau'=(s_0,{s'}_1,{s'}_2)$ that have the same terminal energies $\mathcal{E}(s_1)=\mathcal{E}(s'_2)=-1$, where the potentials are defined as $\phi _{\theta}({s_0}\rightarrow {s_1})=-1$ and $\phi _{\theta}({s_0} \rightarrow {s'}_1)={\phi _{\theta}}({s'}_1 \rightarrow {s'}_2)=-0.5$. Then, it induces $P_F(s_1|s_0)>P_F({s'}_1|s_0)$ which makes $P^{\top}_F(s_1)\neq P^{\top}_F({s'}_2)$.
> > >
> > > ---
> > >
> > > **Q1-Experiment-1. The number of calls for FL-GFN and GFNs differs, as FL-GFNs need to evaluate all intermediate energies.**
> > >
> > > Yes, as you understand, FL-GFN requires more calls than GFNs, as FL-GFN evaluates true energies for all intermediate states, while GFNs require evaluating the terminal energy.
> > >
> > > ---
> > >
> > > **Additional minor issues.**
> > >
> > > Thank you again for pointing out detailed errors! We update our manuscript to follow your suggestions.
> > >
> > > - We add the description of $F(\cdot)$ in subTB of **Section 2.1**.
> > > - We modify the 'Partial inference for GFlowNets' to 'Partial inference in GFlowNets' in **Section 2.2**.
> > > - We modify $\phi$ to $\phi_{\theta}$.
> > > - As stated in the response of **Q2-1-Method-1**, **Smoothing** of **Figure 10** corresponds to GFlowNets with uniformly redistributed energy decomposition (**Figure 11** is related to the correction methods for the decomposition error).

---

> > > > ### Comment · Reviewer_qpM8 · 2023-11-20
> > > > **Thank you for your response!**
> > > >
> > > > Thank you for your detailed response.
> > > >
> > > > ---
> > > >
> > > > **Q2-1-Method-1**
> > > >
> > > > In terms of my comment, I realized that I pointed at a wrong figure. Figure 10(a) and Figure 10(b) show the comparison between energy decomposition methods. If I want to see GFN-smoothing vs. vanilla GFN, I need to compare 10(a) with 7(a), as well as 10(b) with 5. To conclude, GFN-smoothing < vanilla GFN in the set generation, while GFN-smoothing > vanilla GFN in the molecule generation.
> > > >
> > > > ---
> > > >
> > > > I raised the score: 6 --> 8.

---

### Official Review · Reviewer_RMo5 · 2023-10-31

**Soundness:** 3 good
**Presentation:** 3 good
**Contribution:** 4 excellent
**Rating:** 8
**Confidence:** 5

**Summary:**

A method is proposed to improve training of generative flow networks (GFlowNets) using a learned reward shaping scheme. Specifically, one trains an auxiliary model to predict an energy delta for every edge, with the objective that the sum of energy deltas along any trajectory should equal the negative log-reward of the state at which the trajectory terminates. The learned energy delta is then used as a correction to the log-flow difference that appears in the detailed balance training objective. This method is shown to accelerate convergence and improve mode discovery in several very different problems from past work where GFlowNets have been successfully applied.

**Strengths:**

- Good exposition; I have very few complaints on the presentation and math.
- Natural and well-carried-out idea that seems to substantially improve GFlowNet training.
- Experimental validation in very diverse settings from past work. Comparisons to alternative ways of learning the energy decomposition make the paper stronger.
  - But see questions below.

**Weaknesses:**

- I take issue with the use of "partial inference of GFlowNets" in the title and "partial inference" elsewhere and would strongly suggest for this to be revised.
  - "Inference of X" means, given some information related to X, producing an instance of X or a distribution over X. So "partial inference of GFlowNets" should mean learning part of a GFlowNet, or learning it from incomplete information. This is not what is done in this paper: the whole GFlowNet is learned, but some terms in the objective can be computed using partial trajectories.
  - The third paragraph of the introduction and the second paragraph of 2.2 attributes "partial inference" to [Pan et al., 2023a], but in fact that paper does not introduce this term and does not ever use it.
  - The minimal fix would be to make the title "partial inference ~of~ **in**/**for** GFlowNets", which would more accurately describe what is being done.
- Small errors in exposition on GFlowNets:
  - Error on the top of p.3: $\exp(-{\cal E}(x))$ is the reward, not the energy.
  - End of 2.1: SubTB as written does not "interpolate between DB and TB". It only interpolates if one uses the $\lambda$ parameter from [Madan et al., 2023], in which case it indeed interpolates between DB ($\lambda\to0+$) and TB ($\lambda\to+\infty$).
  - 2.2: As written, (2) simply does not make sense: $\cal E$ is defined to be a function with domain $\cal X$, but then it is applied to nonterminal states $s$! This can be fixed by writing that FL **assumes** an extension of $\cal E$ to nonterminal states, and that the freedom we have in the choice of this extension is a starting point for this paper (+ briefly discuss possible sources of the partial energies).
- Limitations/costs/failure modes of the proposed algorithm are not discussed.
- Missing details on the form of the model that predicts $\phi$ (I could not find them in the appendix). Does it share some parameters with the flow and policy models? It is an interesting question how simple or complex the energy decomposition model needs to be, relative to the policy model, in order to be useful.
- Some theoretical characterization of the optimal energy decomposition would be helpful (even in the case of a tree-shaped state space).
- In all plots with curves, please use markers or line styles, and not just colour, to distinguish the curves.

I am very willing to raise the score to 6 or even 8 if the above weaknesses and below questions are addressed.

**Questions:**

- On alternative potential learning:
  - It is a little surprising to me that learning the $\phi(s\rightarrow s')$ works so much better than learning an extension of $\cal E$ to nonterminal states, just replacing $\phi(s\rightarrow s')$ by ${\cal E}(s')-{\cal E}(s)$ in the current objective (section 4.5). Such an expression automatically guarantees a cycle consistency property for $\phi$.
  - Why do you need to learn a proxy model to predict terminal energies instead of using true rewards?
  - Additionally, one could regress each $\phi(s\to s')$ to $\frac1T{\cal E}(s_T)$ (or, using energies, regress ${\cal E}(s_t)$ to $\frac tT{\cal E}(s_T)$). This would also decrease variance and sparsity.
- I am curious how well the energy decomposition performs with SubTB and how it could be used with TB, which does not require learning state flows (while the current system requires **three** estimators for every edge, in addition to one per state). For example, one could add the predicted energy difference to the logits of the forward policy. Do you have any ideas about this question?

---

> ### Author Response · Authors · 2023-11-15
>
> Dear reviewer RMo5,
>
> We express our deep appreciation for your time and insightful comments. In our updated manuscript, we highlight the changes in $\color{blue}{\text{blue}}$.
>
> In what follows, we address your comments one by one.
>
> ---
>
> **W1-1/W1-3. Rather than "partial inference of GFlowNets", "partial inference in/for GFlowNets" is more proper for what is being done in this paper.**
>
> We appreciate your detailed comments to improve our paper! In our updated manuscript, we incorporate your valuable suggestion by modifying "partial inference of GFlowNets" to "partial inference in GFlowNets".
>
> ---
>
> **W1-2. Section 2.2 attributes "partial inference" to FL-GFN [1], but that paper does not introduce this term.**
>
> We would like to clarify our terminology stems from FL-GFN [1] which states "partial inferences that do not contain a full specification of the latent variables that explain the given input" in the last sentence of paragraph six of the introduction section. To alleviate your concern, we modify our description to be more precise by stating "Pan et al. (2023a) first incorporate this concept for training GFlowNets".
>
> ---
>
> **W2. There are small errors in the exposition on GFlowNets.**
>
> Thank you for your insightful comments! We fix **Section 2** by following all of your detailed comments.
>
> - In **Detailed balance** of **Section 2.1**, we fix an error at the top of p.3 (which states $\exp(-\mathcal{E}(x))$ as the energy) by modifying it as the "exponent of the negative energy, i.e., the score of the object".
> - In **Sub-trajectory balance** of **Section 2.1**, we clarify that we actually consider subTB$(\lambda)$ which interpolates between the DB and the TB.
> - In **Forward-Looking GFlowNet** of **Section 2.2**, we clarify that FL assumes an extension of $\mathcal{E}$ to nonterminal states.
>
> ---
>
> **W3. Limitations/costs/failure modes of the proposed algorithm are not discussed.**
>
> We incorporate your valuable comments in our new discussion section (**Appendix D**).
>
> As limitations, our method requires additional time and space costs to learn a potential function $\phi_{\theta}$. However, the increment is minor for real-world problems with prohibitively expensive energy evaluation, such as the drug discovery task that requires evaluating with docking tools or wet-lab experiments.
>
> For detailed costs, we report the additional time costs of LED-GFN by comparing with the GFlowNets. In **Appendix E.2** of our updated manuscript, we (1) provide time costs in molecular generation task, where the conventional subTB objective takes $5.80$ seconds for obtaining $100$ samples during training while our LED-GFN takes $6.61$ seconds, and (2) discuss the efficiency with respect to the time costs.
>
> As failure modes, one can consider an extreme case where most actions are meaningless, i.e., they do not affect energies or the terminal state.  In this case, the smoothed local credits are not meaningful. This extreme case can be avoided with well-designed actions and state spaces.
>
>
> ---
>
> **W4-1. What is the form of the model $\phi_{\theta}$? Does it share some parameters with the flow and policy models?**
>
> In all experiments, we design the potential functions $\phi_{\theta}$ to have the same architecture as the flow model, e.g., a feedforward network with the same hidden dimensions, where the input dimension is extended to consider the transition $s_t \rightarrow s_{t+1}$. We modify the first sentence of **Appendix C** to better highlight these points. Additionally, the potential model $\phi_{\theta}$ does not share any parameters with the flow and policy models.
>
>
>
> ---
>
> **W4-2. In order to be useful, how simple or complex the energy decomposition model needs to be, relative to the policy model?**
>
> Empirically, we find that the complexity of the energy decomposition model is sufficient when its complexity matches that of the flow (or policy) models. Intuitively, one can conclude that estimating the decomposition of an energy function is as challenging as learning the energy function itself, which is also necessary for the flow model.
>
> ---
>
> **W5. Some theoretical characterization of the optimal energy decomposition would be helpful.**
>
> Indeed, our work lacks theoretical aspects since we mainly focus on improving the performance of GFlowNets in practice.
>
> As our best effort to incorporate your comments, we provide the theoretical characterization of the optimal energy decomposition in **Appendix F** of our new manuscript. To be specific, we derive the necessary conditions for the potential functions to perfectly minimize the energy decomposition loss. The result implies a positive correlation between a potential function evaluated on a state transition and the energies of the terminal states reachable from the state transition. This hints at the meaningful-ness of the local credits provided by the potential function.

---

> ### Author Response · Authors · 2023-11-15
>
> **W6. Plots with different markers or line styles would be helpful.**
>
> Thank you for your valuable suggestions! We modify all plots to incorporate different markers or line styles in our updated manuscript.
>
> ---
>
> **Q1-1. On alternative potential learning (Figure 10), why learning $\phi(s\rightarrow s')$ (ours) is better than learning an extension of $\mathcal{E}$ to nonterminal states (simple model)?**
>
> We first clarify that the 'simple model' in **Figure 10** uses a model trained to predict the terminal energy from the terminal state but not trained with the intermediate states. Here, we assume that learning $\phi(s\rightarrow s')$ has shown better results, as the 'simple model' may return not informative values for intermediate states that are not considered during training.
>
> ---
>
> **Q1-2. On alternative potential learning (Figure 10), why do you need to learn a proxy model to predict terminal energies (simple model) instead of using the true energy function?**
>
> We consider such a baseline since it can be a solution to address one of our motivations: true energy evaluation for intermediate states can be expensive (described in **Section 3.1**). Since this baseline can predict the intermediate state energy as a model-based approach, there is no need to evaluate the true energy for the intermediate state.
>
> ---
>
> **Q1-3. Why do you not regress each $\phi(s\rightarrow s')$ to $\frac{1}{T}\mathcal{E}(s_T)$?**
>
> We do not set $\phi_{\theta}(s \rightarrow s') = \mathcal{E}(s_T)/T$ for a specific trajectory since the potential is associated with *multiple trajectories* $\\{\tau | (s\rightarrow s') \in \tau\\}$. In other words, the potential $\phi_{\theta}(s \rightarrow s')$ should be designed to consider:
> \begin{equation*}
>     \mathcal{E}(s_{T})\approx\sum_{(s_{t}, s_{t+1}) \in \tau}\phi_{\theta}(s_{t}\rightarrow s_{t+1})\quad \text{for all } \tau=(s_{0},\ldots,s_{T}) \in \\{ \tau | (s\rightarrow s') \in \tau \\}.
> \end{equation*}
> Then, one can observe that the condition $\phi_{\theta}(s \rightarrow s') = \mathcal{E}(s_T)/T$ for a specific trajectory would not be optimal for another trajectory. In **Figure 10** of our updated manuscript, we add such a baseline, i.e., set $\phi_{\theta}(s \rightarrow s') = \mathcal{E}(s_T)/T$ for a given trajectory at each step. One can see that our learning method still exhibits the most competitive performance.
>
> ---
>
> **Q2. Can you apply your approach to the TB-based objective, e.g., reparameterize logits in the policy?**
>
> Thank you for the interesting suggestion! We follow your valuable suggestion by conducting experiments with modified TB that reparameterizes the logits of the policy with the potential. We incorporate corresponding results in **Appendix E.3** of our updated manuscript. One can see that our LED-GFN with your suggested approach improves the TB. We assume that the local credits support assigning a high probability to the action responsible for the low terminal energy.
>
> ---
>
> **Reference**
>
> [1] Pan et al., Better Training of GFlowNets with Local Credit and Incomplete Trajectories, ICML 2023

---

> > ### Comment · Reviewer_RMo5 · 2023-11-18
> > **Thank you for the response**
> >
> > Thank you for the detailed response. I appreciate all the efforts to improve the paper and the new experiments!
> >
> > I raised the score, but I have two comments on the new theoretical analysis in Appendix F:
> > - There is a little error, which is that the analysis assumes all trajectories are of the same length $T$ (when in fact $T$ depends on $\tau$). So it does not make sense to use $T$ outside of an expression where $\tau$ is a bound variable (like a summation over $\tau$).
> > - The analysis is interesting but not fully satisfying, since it just characterizes the optimal $\phi$ as the solution of a linear system with complicated coefficients. Is a more thorough analysis in a restricted setting possible? For instance, at least in a binary tree with fixed depth, I would conjecture that the optimal $\phi$ recover the deltas of optimal log-flows.
> >
> > (Not essential since the main contribution is, as you say, empirical, but interesting to think about!)

---

> ### Author Response · Authors · 2023-11-20
>
> Dear reviewer RMo5,
>
> Thank you very much for the positive response. We think your comments were very helpful in improving our paper, and we are happy to hear that our efforts have addressed most of your concerns!
>
> For additional comments, we agree that "new theoretical analysis lacks a clear explanation (but the theory is not essential for our paper)". Therefore, instead of incorporating the current analysis in the manuscript, we will consider the suggested analysis in our future manuscript as possible, while leaving the in-depth analysis as a future work (stated in the last sentence of **Section 5** of the updated manuscript).

---

### Official Review · Reviewer_q5Nd · 2023-11-01

**Soundness:** 3 good
**Presentation:** 3 good
**Contribution:** 3 good
**Rating:** 8
**Confidence:** 2

**Summary:**

This paper proposes to improve GFlowNet training by assigning rewards or credits to the intermediate steps. Previous work like forward-looking(FL) GFlowNet propose to reparametrize the state flow function and make use of the partial reward. However, this evaluation of the reward of intermediate state can be expensive. Therefore, the author proposes an interesting learnable energy function on state transitions and obtain good results in some tasks.

**Strengths:**

This paper targets at an interesting and valuable question. Both empirical and theoretical results seem solid and convincing. By learning decomposition of the reward in the terminal stage using potential functions, it can improve both detail-balance based or sub-trajectory balance based GFlownet training.

**Weaknesses:**

See questions.

**Questions:**

The author(s) use two energy-based decomposition schemes as a base-line, namely, the model-based GFlowNet (Jain et al.) and LSTM-based decomposition (Arjona-Medina et al.). Why not test other base-lines? It seems like a small comparison, and I’m not entirely sure if the LSTM is comparable to a GFlowNet based model given its simplicity.  Another question would be the paper discussed both DB-based and subTB based objectives. What about other objectives?

---

> ### Author Response · Authors · 2023-11-15
>
> Dear reviewer q5Nd,
>
> We express our deep appreciation for your time and insightful comments. In our updated manuscript, we highlight the changes in $\color{blue}{\text{blue}}$.
>
> In what follows, we address your comments one by one.
>
>
> ---
>
> **Q1-1. In Figure 10, is LSTM comparable to a GFlowNet-based model given its simplicity?**
>
> The 'LSTM' baseline in **Figure 10** is comparable since it is also an energy decomposition-based GFlowNet model. It just uses the LSTM model to predict the potential $\phi_{\theta}(s\rightarrow s')$ to enable partial inference in GFlowNets.
>
> Additionally, we would like to reclarify that **Figure 10** shows the comparison with alternative energy decomposition methods *for partial inference in GFlowNets*, i.e., alternative methods to predict the potential $\phi_{\theta}(s\rightarrow s')$ for GFlowNets. In our updated manuscript, we modify the description of corresponding ablation studies (last paragraph of **Section 4.5**) to better reflect this point!
>
>
> ---
>
> **Q1-2. Why not test other baselines for energy decompositions?**
>
> We do not conduct extensive comparisons between energy decomposition methods since (1) there are no prior studies about energy decomposition-based GFlowNets and (2) our main focus is first to show the effectiveness of energy decomposition in GFlowNets.
>
> Nevertheless, to further address your comment, we add an additional energy decomposition baseline by modifying a reward redistribution method in reinforcement learning [1] to decompose the energy for GFlowNets. In **Figure 10** of our updated manuscript, one can see that our method still shows the most competitive performance.
>
> If you provide information about an essential baseline, we would be happy to incorporate it!
>
> ---
>
> **Q2. Why not consider the other objectives beyond DB-based and subTB-based objectives?**
>
> The other conventional objective, i.e., TB, is not considered since TB is defined with the full trajectory but our goal is to enable partial inference for the local trajectory-based objectives, i.e., DB and subTB. However, it is worth noting that our new experiments in **Appendix E.3** show that our energy decomposition even improves the TB-based objectives. For other objectives extending DB or subTB [2], combining LED-GFN with them can be an interesting future research direction.
>
>
> ---
>
> **Reference**
>
> [1] Gangwani et al., Learning Guidance Rewards with Trajectory-space Smoothing, NeurIPS 2020
>
> [2] Pan et al., Generative Augmented Flow Networks, ICLR 2023

---

### Official Review · Reviewer_xgzq · 2023-11-01

**Soundness:** 3 good
**Presentation:** 3 good
**Contribution:** 3 good
**Rating:** 8
**Confidence:** 3

**Summary:**

The paper studies the problem of credit assignment in the context of GFlowNets. GFlowNets are amortized samplers which are trained to sample from a target energy function. Typically, GFlowets are only trained using this energy as the terminal reward for the generated object.  Even cases where some intermediate energy signals are available, using them can be computationally expensive. Similar challenges are also studied in the context of reinforcement learning in environments with sparse rewards, which includes approaches such as RUDDER which models the decomposition of returns. This paper proposed LED-GFN a method which learns a decomposition of the energy function to enable partial inference in the context of GFlowNets. The method leverages the flow reparameterization from forward-looking GFlowNets to utilize these partial energy estimates. LED-GFN decomposes the terminal state energy into learnable potential functions defined on state transitions which serve as local credit signals. The potentials are trained to approximate the terminal energy through summation and minimize variance along the action sequence. This provides dense and informative training signals. LED-GFN is evaluated on set generation, bag generation, molecule generation, RNA sequence generation, and maximum independent set problems. It outperforms GFlowNet baselines which do not use the  and achieves similar performance to methods using ideal intermediate energies.

**Strengths:**

* The paper proposes an interesting approach to tackle the problem of credit assignment and partial inference in GFlowNets. It builds upon the ability of forward-looking GFlowNets and addresses the limitation of having an intermediate potential function by learning it.
* The learned potential function can also act as an important inductive bias during training: approximating energy through summation and minimizing variance over the trajectory can make the energy landscape easier to model for the sampler.
* The method enjoys quite strong empirical performance over baselines on a diverse set of fairly complicated tasks.
* The experiments in the ablations are quite thorough and well designed and provide interesting insights in the method's performance.
* Reproducibility: The authors provide code to reproduce the results for the molecule generation experiments and includes most details to reproduce the results.

**Weaknesses:**

* Learning the decomposition of the terminal potentials is interesting, but there are no theoretical guarantees that the learned decompositon provides meaningful local credits in all settings.
* In terms of the empirical results, while the method performs quite well on a variety of tasks - there is an important caveat to note which is the size of the problems. The trajectory length in the problems considered is quite small. There are no experiments on problems with long trajectories which is where the local credit assginment would be critical and thus demonstrate the efficacy of the approach.
* Another motivation for the approach even in the presence of ideal intermediate signals is that the true intermediate energy function can be expensive to compute. However, all the experiments consider tasks where this is not the case. So it is unclear whether there is a significant computational advantage.
* Another important caveat of the empirical analysis is that it focuses on discrete problems and does not consider the continous case.

**Questions:**

* Can you comment on the scalability of the approach and how much of the benefit does it provide for longer trajectories?
* Have you considered combining the LED-GFN approach in an active learning setting where the terminal reward is expensive to compute and there is no intermediate energy signal? (This seems to be the actual useful case in the molecule generation case)

---

> ### Author Response · Authors · 2023-11-15
>
> Dear reviewer xgzq,
>
> We express our deep appreciation for your time and insightful comments. In our updated manuscript, we highlight the changes in $\color{blue}{\text{blue}}$.
>
> In what follows, we address your comments one by one.
>
> ---
>
> **W1. There are no theoretical guarantees that the learned decomposition provides meaningful local credits in all settings.**
>
> Indeed, since our main focus is on improving the performance of GFlowNets in practice, our work lacks a theoretical guarantee. We provide our best efforts to alleviate your concerns in **Appendix F** of the updated manuscript and what follows.
>
> To this end, we derive a fixed form condition for the potential functions that imply the meaningful-ness of our local credit via the correlation between potentials and the terminal energy (similar to [1]). To be specific, in **Appendix F**, we derive
> \begin{equation*}
> \phi_{\theta}(s\rightarrow s') = K_{1}\sum_{\tau \ni (s\rightarrow s')}\mathcal{E}(s_{T}) + K_{2},
> \end{equation*}
> where the summation over $\tau$ is defined on trajectories that include $s\rightarrow s'$ as its intermediate transition and $s_{T}$ is the terminal state of the associated trajectory $\tau$. Furthermore, $K_{1} > 0$ and $K_{2}$ are constants with respect to the change in $\phi_{\theta}(s\rightarrow s')$ and $\mathcal{E}(s_{T})$.
>
> Our result hints at how the potential function is likely to be positively correlated with the terminal energies $\mathcal{E}(s_{T})$ reachable after the transition $s\rightarrow s'$. This implies how the potential function provides meaningful local credit, as maximizing the potential function likely leads to the maximization of the terminal energy.
>
>
>
> ---
>
> **W2. There are no experiments on problems with long trajectories.**
>
> To alleviate your concern, we conduct additional experiments with relatively long trajectories: the set generation with the size of $80$. We incorporate this experiment as an additional ablation study in **Appendix E.1** in our updated manuscript. Again, one can observe how LED-GFN significantly improves the GFN.
>
>
> ---
>
> **W3. The experiments do not consider tasks requiring expensive energy functions, so the significant computational advantage of LED-GFN is unclear.**
>
> To alleviate your concern, we provide additional experiments in **Appendix E.2** of the updated manuscript that demonstrates a significant computational advantage of LED-GFN even when using moderately expensive energy functions (oracle used in molecular generation).
>
> We compare (1) the detailed time costs and (2) the performance with respect to the time costs with the considered baselines, i.e., GFN and FL-GFN, in molecular generation. Our method (1) requires approximately $14\\%$ of the additional time required for FL-GFN to enable partial inference in GFlowNets, and (2) shows the time-efficient performances compared to the considered baselines.
>
> ---
>
> **W4. Why not consider a task to generate continuous data?**
>
> We focus on discrete objects due to their importance in applications such as drug discovery and protein design. Nevertheless, our framework can be naturally extended to continuous GFlowNets based on detailed balance and sub-trajectory balance objectives. This would be an interesting direction for future research.
>
>
> ---
>
> **Q1. Can you comment on the (1) scalability and (2) how much of the benefits LED-GFN provides for longer trajectories?**
>
> First, our LED-GFN introduces a small overhead compared to the GFlowNet training procedure, as it requires the evaluation of the potential function for each trajectory. In **Appendix E.2**, we provide the detailed time costs in molecular generation tasks. The conventional subTB objective takes $5.80$ seconds to obtain $100$ samples during training while our LED-GFN takes $6.61$ seconds.
>
> Next, longer trajectories require deciding a large number of actions, which is more likely to make the local credit assignment more challenging for GFlowNets. The benefits of our LED-GFN are more significant in these scenarios by providing informative local credits for partial inference in GFlowNets, as supported by our new experiment in **Appendix E.1**.
>
>
> ---
>
> **Q2. Have you considered active learning with expensive energy evaluation and there is no intermediate energy signal?**
>
> Indeed, we did not conduct experiments with expensive energy evaluations, e.g., AutoDock. Nevertheless, we agree that the suggested experiments are interesting to highlight the benefits of our method: the efficiency with respect to the number of samples or energy evaluation. Unfortunately, we are unable to conduct such experiments during the rebuttal phase due to the expensive energy evaluation itself. We will consider this as a future work for evaluating our framework.
>
>
> ---
>
> **Reference**
>
> [1] Arumugam et al., An Information-Theoretic Perspective on Credit Assignment in Reinforcement Learning, NeurIPS 2020 Workshop on Biological and Artificial Reinforcement Learning

---

> > ### Comment · Reviewer_xgzq · 2023-11-18
> > **Response to rebuttal**
> >
> > Thanks for the response!
> >
> > W1:
> > Thanks for the analysis. However, I find this analysis somewhat irrelevant. What the analysis says is *at convergence*, there could be positive correlation between the terminal energy and the learned potential. The important question, however, is what happens in the typical scenario when the loss doesn't go to 0. This is crucial since the trajectories that the energy decomposition is trained on is not IID, and thus there is no guarantee for convergence. Moreover, the connection to [1] isn't clear to me. To be clear, I think the empirical performance is great, but this analysis, in my opinion doesn't add much to the paper. It is possible that I am missing some details so I will be happy to reconsider my comment.
> >
> > W2:
> > Thanks for including this experiment! The method holds an advantage over a vanilla GFlowNet even with longer trajectories. But it is worth noting that the gap with FL-GFN vanishes in the longer tasks. This perhaps points to the possibility that learning the potential gets harder as trajectories get longer, which would make the improved credit assignment claims somewhat weaker.
> >
> > W3:
> > Thanks for sharing the computation time results. The results seem quite promising and presumably the advantage will grow with the length of the trajectories. However, I would like to note that the reward in the molecule generation experiments is still just a neural network which is still computationally cheap.
> >
> > Q1:
> > I believe the experiments on the longer trajectories do show promising results but equally importantly they also point to a potential limitation w.r.t FL-GFN.

---

> ### Author Response · Authors · 2023-11-20
>
> Dear reviewer xgxq,
>
> Thank you for the response! We appreciate your comments to improve our paper.
>
> We would like to clarify the following points to address your additional comments.
>
> ---
>
> **W1-1. The empirical performance is great, but the new theoretical analysis (Appendix F) is not clear, e.g., can not guarantee convergence.**
>
> We agree that our new analysis lacks a clear explanation due to the challenge of guaranteeing optimal conditions in all settings. However, as stated in **response of W1**, it is worth noting that our main focus is on improving empirical performance. Therefore, instead of incorporating the current analysis in the manuscript, we leave in-depth analysis as an interesting future work (stated in the last sentence of **Section 5** of the updated manuscript).
>
> ---
>
> **W2-1/Q-1. In the new experiments for large set generation, LED-GFN shows similar performance compared to the FL-GFN, so the claim of LED-GFN seems to be weakened.**
>
> We would like to clarify that a large set generation is *idealized setting* for FL-GFN as a synthetic task, where the energy function is designed to perfectly identify the contribution of each action, i.e., *ideal local credits* (setting of the set generation discussed in **Section 4.4**).
>
> In this task, achieving performance similar to FL-GFN still supports our claim by highlighting that the learned potentials can be as informative as *ideal local credits* (as extended results of **Figure 7(a)** for longer trajectories). We incorporate this clarification in **Appendix E.1** of the updated manuscript.
>
>
> ---
>
> **W3-1. The advantage of time costs is promising, but experiments still do not consider expensive energy evaluation.**
>
> We are happy to hear that our new experiments address your concerns regarding the advantage of time costs!
>
> For expensive energy evaluation, e.g., docking simulation, we do not consider such a setting which is unable to be conducted during the rebuttal phase due to the expensive costs itself, but we believe that this can be an interesting future direction (as stated in **response of Q2**).

---

> > ### Comment · Reviewer_xgzq · 2023-11-21
> >
> > Thanks again for the clarifications. I appreciate the authors efforts in improving the paper during the rebuttal. I have raised my score accordingly.

---

### Meta-Review · Area_Chair_5GbN · 2023-12-12

**Metareview:**

This paper received a unanimous and strong positive reviews. There were a number of issues raised during initial reviews including - experiments being restricted to small trajectories/ settings with easy-to-calculate local energy functions, comparison to related methods, broader applicability of the ideas among others. All of these have been addressed satisfactorily in the author response. The main unaddressed issues are related to lack of theoretical justifications -- in my view this is not a severe shortcoming considering the empirical nature of the paper and use cases.

**Justification For Why Not Higher Score:**

N/A

**Justification For Why Not Lower Score:**

This is a paper with a new, simple, and efficient methodology to improve sampling from energy distributions using GFlowNets. While I am not an expert on this topic, given the unanimous positive reviews and the reviewer comments on the effectiveness, I would like to recommend oral for the paper.

---

### Decision · Program_Chairs · 2024-01-16

Accept (oral)